# Mitochondrial genomes and phylogeny of Atratus and Educator Group species of the Melanoconion Section of *Culex* (*Melanoconion*) (Diptera: Culicidae)

**Tatiane Marques Porangaba de Oliveira**[1/+], **Peter Gordon Foster**[2],
**Ivy Luizi Rodrigues de Sá**[1], **Maria Anice Mureb Sallum**[1]

[1]Universidade de São Paulo, Faculdade de Saúde Pública, Departamento de Epidemiologia, São Paulo, SP, Brasil
[2]Natural History Museum, Department of Life Sciences, London, United Kingdom

**BACKGROUND** *Culex* (*Melanoconion*) species are known to act as vectors for different arboviruses, and little is known about the mitochondrial genome of these species.

**OBJECTIVES** Aiming to expand the genetic knowledge of this subgenus, a 12Kb fragment of the mitochondrial genome was sequenced from 23 specimens belonging to the Atratus and Educator Groups of the subgenus *Melanoconion* of *Culex*.

**METHODS** The sequenced specimens were morphologically identified as *Culex dunni*, *Culex ensiformis*, *Culex theobaldi*, *Culex trigeminatus*, *Culex eknomios*, *Culex zeteki*, *Culex* near *commevynensis*, *Culex angularis*, *Culex longistriatus*, and *Culex* near *vaxus*. The reads were assembled with the reference genome of *Culex quinquefasciatus* and MITOS2 was used for gene annotation. Values of guanine-cytosine (GC) and adenine-thymine (AT) skews, nucleotide diversity, ratio between non-synonymous (Ka) and synonymous (Ks) substitution, and nucleotide composition were calculated, and phylogenetic analysis was performed.

**FINDINGS** As in other *Culex* mitogenomes, the partial mitochondrial genomes include 12 protein coding genes (PCGs), 15 tRNA, and 1 rRNA (*rrnL*). The PCGs showed no length variation between the species studied. *ND5* gene presented one less base in the new sequences, which generated a stop codon and, consequently, a shorter length in relation to the *Cx. quinquefasciatus* sequence reference. All specimens showed a positive value for AT-skew and negative for GC-skew. Nucleotide diversity varied between 0.00407 and 0.12085. Ka/Ks values ranged from 0.0 to 2.775.

**MAIN CONCLUSIONS** Leucine and Serine were the most abundant amino acids. Phylogenetic analysis suggested three putative species.

Key words: *Culex - Melanoconion* - mitochondrial - genome

The majority of species of the subgenus *Melanoconion* of *Culex* occurs in neotropical regions. Some of the species from this subgenus are recognized for acting as important vectors of several arboviruses affecting humans and other vertebrates, such as Venezuelan Equine Encephalitis complex, West Nile Virus, and Eastern Equine Encephalitis Virus.[1-6] Recently, Agua Salud alphavirus was isolated from *Culex* (*Melanoconion*) mosquitoes collected from the Brazilian Amazon (Belém municipality, Pará State),[7] suggesting that they are natural hosts of these viruses.

Torres-Gutierrez and Sallum[8] updated the catalogue of the subgenus *Melanoconion* published in 1992 by Pecor et al.[9] In the new catalogue, this subgenus comprises two sections, 21 groups, 23 subgroups, and 160 valid species. Recently, revisions were carried out on the Atratus and Educator Groups of the Melanoconion Section of the subgenus *Melanoconion* resulting in the description of eight new species and removal of five species from synonymy.[10,11] Thus, after the revisions this subgenus encompasses 173 valid species[8,10,11] which are present in the Melanoconion and Spissipes Sections.[8,10,11,12,13] The Melanoconion Section includes 150 species divided in 13 groups.[8,10,11] The Atratus Group encompass the following species: *Culex atratus*; *Culex caribeanus*; *Culex commevynensis*; *Culex columnaris*; *Culex comptus*; *Culex dunni*; *Culex ensiformis*; *Culex exedrus*; *Culex longisetosus*; *Culex longistylus*; *Culex loturus*; *Culex spinifer*; *Cx. trigeminatus*; and *Culex zeteki*.[10] The Educator Group includes the species *Culex aneles*; *Culex angularis*; *Culex apeteticus*; *Culex aphyllus*; *Culex bibulus*; *Culex cristovaoi*; *Culex educator*; *Culex eknomios*; *Culex inadmirabilis*; *Culex longistriatus*; *Culex rachoui*; *Culex spiniformis*; *Culex theobaldi*; and *Culex vaxus*.[11]

Taxonomic identification of species of this subgenus by female characters is hampered by the great similarity between these characteristics and, therefore, the identification is based mainly on the morphological identification of male genitalia characters.[10,11,12,13] DNA-based species identification is an alternative and can be used as a complementary tool. Torres-Gutierrez et al.[14] verified that the 658 bp region of subunit I of the cytochrome *c*

Financial support: FAPESP, CNPq (grants 2014/26229-7 and 303382/2022-8, respectively, to MAMS).
+ Corresponding author: porangaba@usp.br | https://orcid.org/0000-0003-0069-6848

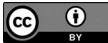

oxidase gene (DNA barcode)[15] presented high resolution in the species delimitation of the subgenus *Melanoconion*. Talaga and Gendrin[16] described three species of *Culex* (*Mel.*) based on both morphological characteristics of the male genitalia and molecular data of the *COI* gene. This mitochondrial marker is also effective in identifying cryptic *Culex* species.[17] Demari-Silva et al.[18] used a fragment of the *COI* gene to establish phylogenetic relationships among 17 species of the genus *Culex*. Although the results corroborate the monophyly of the subgenus *Melanoconion*, the authors suggested confirmation through further studies with nuclear genes and a greater number of samples, including species from the Pilosus Group and Spissipes Section. Torres-Gutierrez et al.[19] verified the monophyly of the Melanoconion and Spissipes Sections through phylogenetic analyses with nuclear and mitochondrial genes. Although the results have been consistent with most of the morphological classification of Spissipes Section, the same was not true for Melanoconion Section. Only the Atratus and Pilosus Groups were monophyletic. Therefore, the authors report that the results for Melanoconion Section were inconclusive due to limited taxon representation and suggest future investigations with greater representation.

Mitochondrial genome has also been used to phylogeny in Culicidae.[20,21] Demari-Silva et al.[20] performed phylogenetic analysis with mitochondrial protein-coding genes using four of the six species of the Coronator Group of the subgenus *Culex* and observed the monophyly of this Group, corroborating with the morphological hypothesis. Mitochondrial genomes in *Culex* corroborates that of other metazoan organisms, presenting a length of approximately 15 Kb, with 13 protein coding genes (PCGs), two rRNA (*rrnL* and *rrnS*), 22 tRNA genes, and a region rich in adenine (A) and thymine (T).[20,21] Mitogenome sequences of *Culex* mosquitoes of the subgenera *Culex*; *Lophoceraomyia*; *Neoculex*; and *Culiciomyia* are available in National Centre for Biotecnology Information (NCBI) (https://www.ncbi.nlm.nih.gov/). Although mosquitoes of the subgenus *Melanoconion* are of medical importance and are used in phylogeny and taxonomy studies,[14,19] information about the mitochondrial genome is scant. Phylogenetic analyses with mitochondrial protein-coding genes together with morphological data can contribute to a more accurate identification of the species of Melanoconion Section. Thus, this study aims to: (1) obtain, describe, and analyse mitochondrial protein coding genes from different species of the Educator and Atratus Groups of the subgenus *Melanoconion*; (2) verify the phylogenetic relationships of both Groups.

## MATERIALS AND METHODS

*Mosquito Sampling* - Mosquitoes of the genus *Culex* subgenus *Melanoconion* from different localities of Brazil were used in this study. Details about mosquito collections are shown in Supplementary data (Table I). Specimens were identified according to the keys proposed by Sá et al.[10] and Rodrigues de Sá et al.,[11] placed in tubes containing 95% ethanol and stored at -80ºC. Male genitalia were dissected and mounted on microscope slide, covered with fine coverslip, and deposited in the Coleção Entomológica de Referência, Faculdade de Saúde Pública, Universidade de São Paulo. Genomic DNA from whole mosquitoes was individually extracted using the Qiagen DNeasy Blood & Tissue Kit (Qiagen), following the manufacturer's instructions. The extracted DNA was stored at -80ºC as part of the frozen entomological collection of the Faculdade de Saúde Pública, Universidade de São Paulo, Brazil.

*Polymerase chain reaction (PCR) amplification and sequencing* - The largest region of the mitochondrial genome between the 16S rRNA and cytochrome *c* oxidase subunit I (*COX1*) genes was amplified with the primers 16Sa (5′ CGCCTGTTTATCAAAAACAT 3′)[22] and LCO1490 (5′ GGTCAACAAATCATAAAGATATTGG 3′).[23] This region encompasses 12 PCGs and has approximately 12 Kb. For each polymerase reaction was used GoTaq® Long PCR Master Mix 1x, 0.2 mM of each primer, 1 µL of DNA and ultrapure water to the final volume of 50 µL. Thermal cycler conditions were 94ºC for 2 min, 42 cycles of 94ºC for 30 s, 45ºC for 20 s and 65ºC for 13 min and a final extension at 72ºC for 7 min. Amplicons were purified using DNA Clean & Concentrador™ (Zymo Research, California, USA) and quantified using a Qubit 2.0 fluorometer (LifeTechnologies, Oregon, USA), according to the manufacturer's instructions. Long PCR amplicon libraries were prepared using the Nextera® XT DNA Sample Preparation Kit (Illumina, Illinois, USA) and paired-end fragments (150 bp) were sequenced on the Illumina MiSeq platform. The quality of the generated reads was assessed using FastQC v0.11.9.[24]

*Mitochondrial genome assembly and annotation* - Mapping to reference method was used to genome assembly. Geneious Prime 2023.2.1 (https://www.geneious.com) was used for paired-end reads assembly using default parameters, mapper Geneious method and *Culex quinquefasciatus* mitochondrial genome sequence (Genbank accession NC_014574) as reference. Annotation of the genes was performed using MITOS 2 Web Server[25] with invertebrate genetic code and confirmed manually with the alignment of each gene with *Culex* sequences available in Genbank (Genbank accessions NC_036006 and NC_014574).

*Sequence analysis* - All new sequences were aligned using ClustalW in MEGA 11.0.13 software.[26] The GC-skews and AT-skews were measured using the following formulas: guanine-cytosine (GC-skews) $(G - C) / (G + C)$ and adenine (A) and thymine (T) (AT-skews) $(A - T) / (A + T)$. These values can range from -1.0 to 1.0 and indicate compositional asymmetries in DNA sequences. Nucleotide diversity (π) was generated in DnaSP v.6.12.0.3,[27] using all mitochondrial sequences obtained and with a sliding window of 200 bp and steps of 25 bp. The ratio between non-synonymous (Ka) and synonymous (Ks) substitution in PCG sequences was also calculated. Values Ka/Ks were obtained from pairwise sequence comparisons of each PCG in DnaSP v.6.12.03. This analysis allows estimating whether certain PCG are under (1) positive selection (Ka/Ks > 1), (2) negative selection (Ka/Ks < 1), or (3) neutral evolution (Ka/Ks = 1).

The invertebrate mitochondrial genetic code was used to translate the PCG sequences in amino acid sequences. Nucleotide composition and relative synonymous codon usage (RSCU) were calculated using MEGA 11.0.13 software.

*Phylogenetic analysis* - An overview phylogeny of Culicidae was made using translations from samples of Culicidae species and the newly-sequenced *Melanoconion* species. Complete mitochondrial genomes from Culicidae were obtained from Genbank, which included 21 genera with at least one mitogenome. Two mitogenomes were sampled randomly from each such genus (with only one from *Orthopodomyia*, as that was all that was available), from which translations of protein coding genes were obtained. New *Melanoconion* sequences PQ389 and PQ409 from this study were added to the other Culicidae sequences. Alignments of the translations were made using Clustalo,[28] but alignments for *ATP8*, *ND3*, and *ND4L* were too short (lengths 54, 118, and 99, respectively) and were not used. A concatenated alignment of the remaining nine translations was made. There were five pairs of sequences that were identical; one of each was removed from the alignment. The alignment of the remaining 38 sequences, with a length of 3137 characters, was analysed with Phylobayes MPI V1.9, using two runs, using the CAT+GTR+G(4) model.[29] A phylogeny of the *Melanoconion* subgenus was made in a similar way to the Culicidae phylogeny described above. Alignments were made using protein-coding gene translations of the 23 new *Melanoconion* sequences from this study, to which were added *Culex* and other near outgroup sequences. This alignment of 46 sequences, of length 3127 AAs, was analysed as described above for Culicidae, using Phylobayes MPI V1.9 using the CAT+GTR+G(8) model.

## RESULTS

*Mosquito sampling* - Of the total specimens used in this study, two were morphologically identified as *Cx. dunni*, two as *Cx. ensiformis*, three as *Cx. theobaldi*, three as *Culex trigeminatus*, two as *Cx. eknomios*, two

as *Cx. zeteki*, one as *Culex* near *commevynenis*, two as *Culex* near *vaxus*, three as *Cx. angularis*, and three as *Cx. longistriatus*. All specimens were male except one *Cx. zeteki* [Supplementary data (Table I)].

*PCR amplification and sequencing* - A region of the mitochondrial genome of approximately 12 kb was amplified from each of the 23 specimens. Next-Generation sequencing generated a total of 2,684,992 paired-end reads. The number of reads generated for each specimen are in Supplementary data (Table II).

*Mitochondrial genome assembly and annotation* - After assembly, the sequences had length between 11,801 and 11,819 base pairs [Supplementary data (Table II)] and the PCGs were composed of 3,392 codons. The partial mitochondrial genome of all samples contains 28 genes, including 12 PCGs (*COX1*, *COX2*, *ATP8*, *ATP6*, *COX3*, *ND3*, *ND5*, *ND4*, *ND4L*, *ND6*, *CYTB*, and *ND1*), 15 tRNA, and 1 rRNA (*rrnL*), with partial sequences of *COX1* and *rrnL* (Fig. 1). Protein coding and tRNA genes were located in both strands (H or L) and *rrnL* on the L-strand (Fig. 1). In H-strand are following PCGs and tRNA: *COX1*, *COX2*, *ATP8*, *ATP6*, *COX3*, *ND3*, *ND6*, *CYTB*, *trnL2*, *trnK*, *trnD*, *trnG*, *trnR*, *trnA*, *trnN*, *trnE*, *trnT*, *trnS*2, therefore in L-strand are: *ND5*, *ND4*, *ND4L*, *ND1*, *trnS1*, *trnF*, *trnH*, *trnP*, *trnL1*. The following nine mitochondrial genes were not sequenced: *ND2* (PCG); *trnI*, *trnQ*, *trnM*, *trnW*, *trnC*, *trnY*, and *trnV* (tRNA); *rrnS* (rRNA).

All PCGs of species sequenced possessed the start codon ATN, except for the *COXI*, which was not possible to determine due to the partial sequencing of this gene (Table S3). The partial stop codon (T_) was observed in *COX1* and *COX2* genes, whereas *ATP6*, *ATP8*, *ND1*, *ND3*, *ND4*, *ND4L*, *ND5*, *ND6*, *COX3* and *CYTB* showed a complete stop codon (TAA) [Supplementary data (Table III)].

The PCGs showed no length variation between the species studied. One difference was observed in the length of the *ND5* gene in relation to the *Cx. quinquefasciatus* sequence (Genbank accession NC_014574), used as a reference for genome assembly. The *ND5* gene of the new sequenced samples presented one less

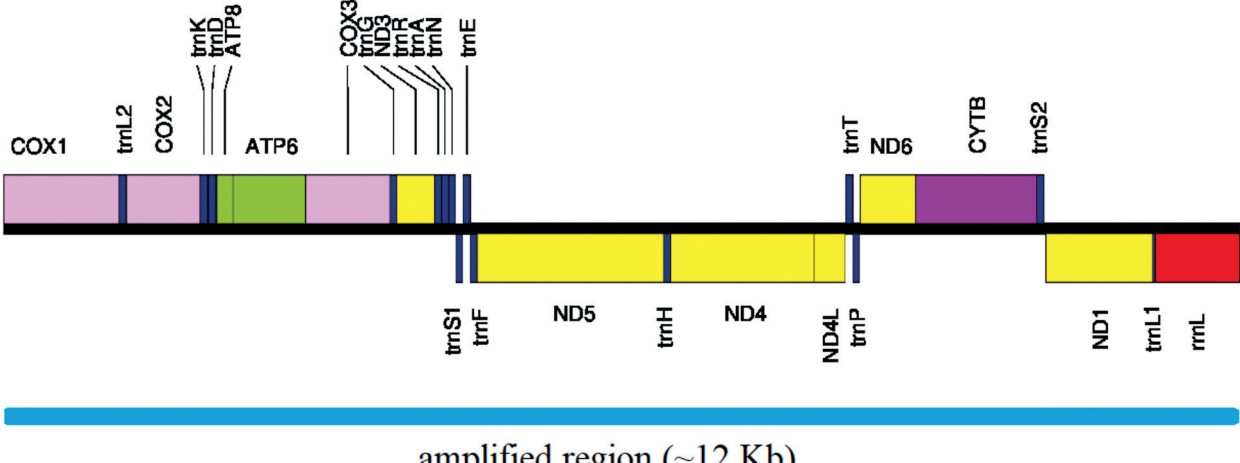

Fig. 1: diagram graphic of the mitochondrial region amplified by the primers mentioned in the study. The region comprises 12 protein coding genes.

base, which generated a stop codon and, consequently, a shorter length compared to the NC_014574 sequence (Fig. 2). Because of the absence of this base occurred in a polyT region, Sanger sequencing was performed to verify whether this difference could have been a result of the assembly of the reads. So, the new sequences were aligned and primers in the *ND3* and *ND5* regions were designed (Primer F: TTGGACTTTATCATGAATG (*ND3* region) and Primer R: GATCAAGGGTGAAGT-GAA (*ND5* region)) to amplify the fragment containing the polyT region mentioned above. Amplification and sequencing of this fragment (~ 622 bp) was performed in the following samples: PQ401 (IG20-5), PQ403 (IG20-6), and PQ408 (SP-Ciduni-02). The result of the alignment of the *ND5* fragment between Sanger sequencing and Illumina sequencing can be seen in Supplementary data (Fig. 1), confirming the absence of the base.

*Sequence analysis* - The nucleotide frequency in the fragment sequenced for each specimen can be seen in Table. All species showed the positive value for AT-skew and negative for GC-skew (Fig. 3), showing an excess of A over T and C over G.

For all species under study, Leu and Ser were the most abundant amino acids and the most frequent codons (RSCU > 1.0) were those ending with A or T, as can be seen for *Cx. dunni* and *Cx.* near *commevynenis* in Fig. 4 and for the other species in Supplementary data (Table IV).

Ka/Ks values ranged from 0.0 to 2.775 for species under study. Most mitochondrial PCGs presented Ka/Ks values lower than 1, indicating under negative selection pressure (Fig. 5). The *ATP8* gene was the one that presented the greatest Ka/Ks variation, being greater than 1 in the following pairwise sequence comparisons: *Cx. longistriatus* (PQ404, PQ390, PQ393) x *Cx. angularis*

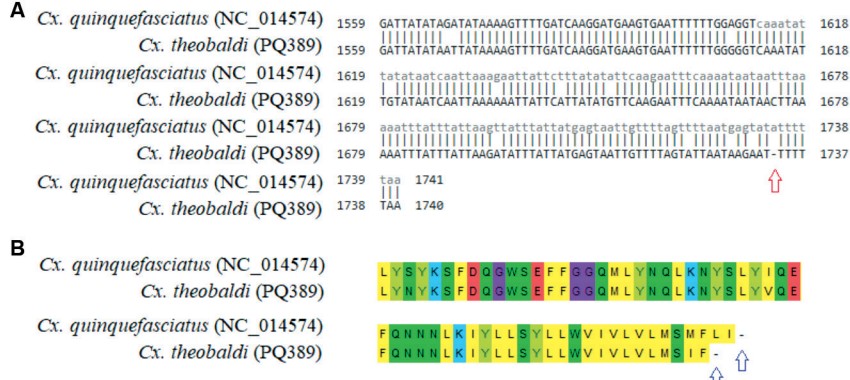

Fig. 2: schematic diagram showing the end of the *ND5* gene nucleotide and amino acid alignment of *Culex quinquefasciatus* (NC_014574) with *Culex theobaldi* (PQ389). (A) Both nucleotide strands are sense direction. The red arrow indicates the lack of a base in PQ389. (B) The blue arrows indicate the stop codon of each specimen for *ND5*.

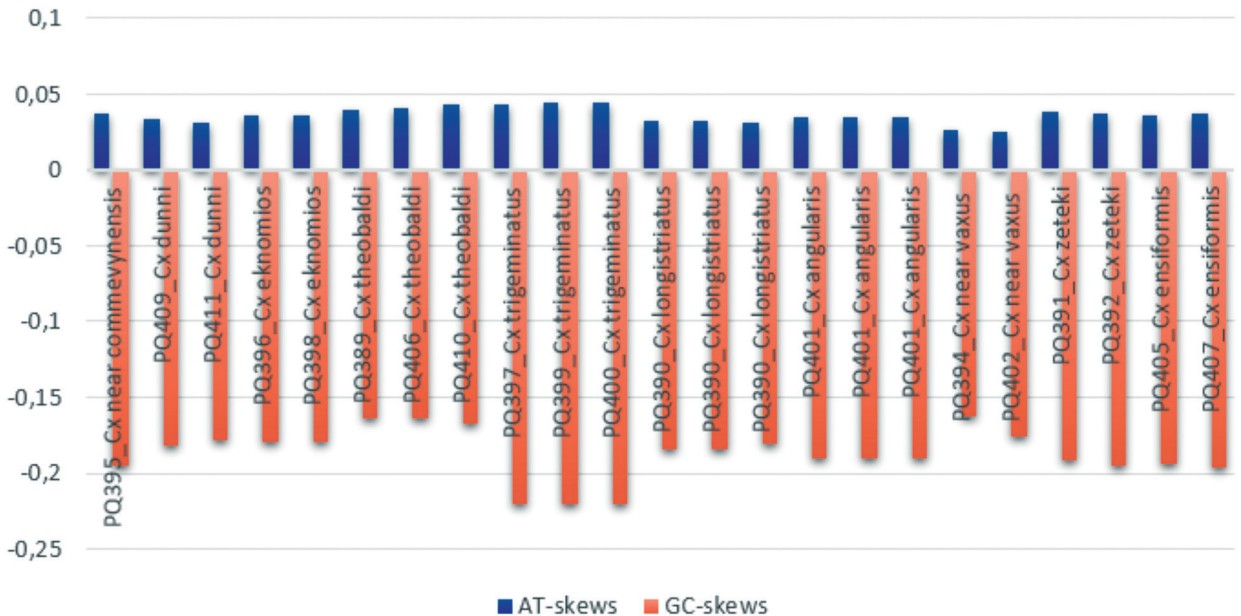

Fig. 3: graph of adenine-thymine (AT) skews and guanine-cytosine (GC) skews values for each species addressed in this study.

TABLE

Nucleotide frequency, AT-skews and GC-skews for each specimen

| ID_sequences | Species | T% | C% | A% | G% | AT-skews | GC-skews |
|---|---|---|---|---|---|---|---|
| PQ395 | *Culex* near *commevynensis* | 36 | 15 | 38,8 | 10,1 | 0,037433 | -0,195219 |
| PQ409 | *Culex dunni* | 36,4 | 14,6 | 38,9 | 10,1 | 0,033201 | -0,182186 |
| PQ411 | *Cx. dunni* | 36,5 | 14,5 | 38,8 | 10,1 | 0,030544 | -0,178862 |
| PQ396 | *Culex eknomios* | 36,9 | 13,8 | 39,6 | 9,6 | 0,035294 | -0,179487 |
| PQ398 | *Cx. eknomios* | 36,9 | 13,8 | 39,6 | 9,6 | 0,035294 | -0,179487 |
| PQ405 | *Culex ensiformis* | 36 | 15,1 | 38,7 | 10,2 | 0,036145 | -0,193676 |
| PQ407 | *Cx. ensiformis* | 35,9 | 15,2 | 38,7 | 10,2 | 0,037534 | -0,196850 |
| PQ389 | *Culex theobaldi* | 36,3 | 14,2 | 39,3 | 10,2 | 0,039683 | -0,163934 |
| PQ406 | *Cx. theobaldi* | 36,3 | 14,2 | 39,4 | 10,2 | 0,040951 | -0,163934 |
| PQ410 | *Cx. theobaldi* | 36,1 | 14,3 | 39,3 | 10,2 | 0,042440 | -0,167347 |
| PQ397 | *Culex trigeminatus* | 35,7 | 15,5 | 38,9 | 9,9 | 0,042895 | -0,220472 |
| PQ399 | *Cx. trigeminatus* | 35,6 | 15,5 | 38,9 | 9,9 | 0,044295 | -0,220472 |
| PQ400 | *Cx. trigeminatus* | 35,6 | 15,5 | 38,9 | 9,9 | 0,044295 | -0,220472 |
| PQ390 | *Culex longistriatus* | 37,1 | 13,8 | 39,6 | 9,5 | 0,032595 | -0,184549 |
| PQ393 | *Cx. longistriatus* | 37,1 | 13,8 | 39,6 | 9,5 | 0,032595 | -0,184549 |
| PQ404 | *Cx. longistriatus* | 37,2 | 13,7 | 39,6 | 9,5 | 0,031250 | -0,181034 |
| PQ401 | *Culex angularis* | 36,8 | 14,1 | 39,4 | 9,6 | 0,034121 | -0,189873 |
| PQ403 | *Cx. angularis* | 36,8 | 14,1 | 39,4 | 9,6 | 0,034121 | -0,189873 |
| PQ408 | *Cx. angularis* | 36,8 | 14,1 | 39,4 | 9,6 | 0,034121 | -0,189873 |
| PQ394 | *Culex* near *vaxus* | 37 | 13,9 | 39 | 10 | 0,026316 | -0,163180 |
| PQ402 | *Cx.* near *vaxus* | 37,4 | 13,7 | 39,3 | 9,6 | 0,024772 | -0,175966 |
| PQ391 | *Culex zeteki* | 36 | 14,9 | 38,9 | 10,1 | 0,038718 | -0,192000 |
| PQ392 | *Cx. zeteki* | 36 | 15 | 38,8 | 10,1 | 0,037433 | -0,195219 |

A: adenine; C%: cytosine; G%: guanine; T: thymine; AT-skew: adenine-thymine skew; GC-skews: guanine and cytosine.

(PQ401, PQ403, PQ408), indicating positive selection pressure [Fig. 5 and Supplementary data (Fig. 2)].

The degree of polymorphism between the different species of *Culex* (*Mel.*) was observed throughout the mitochondrial sequences by nucleotide diversity (π), which varied between 0.00407 and 0.12085 [Supplementary data (Table V)]. The most polymorphic PCGs were *ND6*, *ND5*, and *COX1* (Fig. 6).

*Phylogenetic analysis* - Two phylogenetic analyses were performed using translations from PCGs. Only partial sequence of *COX1* was considered in the analyses, since their complete sequence was not obtained in the 23 sequenced samples under study. The 12kb fragment from the new *Melanoconion* sequences from this study contained *COX1* (partial), *COX2, ATP8, ATP6, COX3, ND3, ND5, ND4, ND4L, ND6, CYTB*, and *ND1*. However, ATP8, ND3, and ND4L were not used because they were, with lengths 53, 117, and 98 AAs respectively, deemed too short. The first tree has a broad phylogenetic range and shows where samples from the *Melanoconion* subgenus fit into the Culicidae family (Fig. 7). The second tree was performed using all 23 *Melanoconion* sequences plus sequences from close groups (Fig. 8). The Genbank acces-

sion number of the sequences used in both phylogenetic trees is indicated after each species name in Figs 7-8.

Phylogenetic analysis generated two well-supported (100% posterior probability) monophyletic clades. A clade with lineages of *Culex* (*Culex*), *Culex* (*Neoculex*), and *Culex* (*Lophoceraomyia*) and one sister clade with *Cx.* (*Mel.*). Within the *Cx.* (*Mel.*) clade, the subclade with species from the Educator Group of the Melanoconion Section was well supported (100%) with the subclade sister of lineages from the Atratus Group of the Melanoconion Section (Fig. 8).

The analysis showed three possible putative species, two (*Culex* near *vaxus* Form 1 and *Culex* near *vaxus* Form 2) resulting from 2 specimens morphologically identified as *Cx.* near *vaxus* and one from the specimen *Cx.* near *commevynensis* (Fig. 8).

## DISCUSSION

Recently, reviews of the Atratus and Educator Group of the Melanoconion Section of *Culex* (*Mel.*) have been published.[10,11] New species were described, species distribution was updated and morphological identification keys for different mosquito life stages were provided, contributing to the identification of species in these Groups.

Fig. 4: relative synonymous codon usage (RSCU) of the mitochondrial partial genome of species of *Culex dunni* and *Culex* near *commevynensis*. The RSCU values can observed on the y-axis.

Species from these Groups are vectors of different parasites to humans, and are therefore important in public health.[30] Previous studies have obtained and analysed mitochondrial and nuclear gene sequences, showing that gene fragments can be used as a tool in the identifying these species.[14,19]

In this study, the mitochondrial sequence of approximately 12 kb was obtained from 23 mosquitoes, comprising eight valid species and three putative species of *Culex* (*Mel.*). This region encompassed 12 PCGs, 15 transfer RNA genes and *rrnL*, with partial sequences of *COX1* and *rrnL*. Although there was no difference in the order of PCGs or tRNAs in relation to other mitochondrial genomes of *Culex*,[20,31] a difference in the length of the *ND5* gene was observed. Variations in mitochondrial DNA length in insects are generally associated with evolutionary events in the control region[32,33,34] and intergenic region.[20] Liu and Beckenbach[35] analysed *COX2*

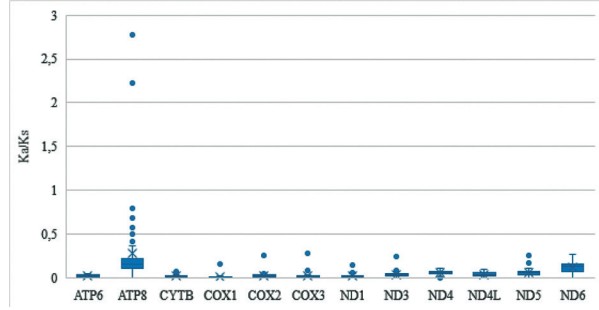

Fig. 5: box plot of the Ka/Ks values of each protein coding gene of *Culex* (*Melanoconion*) species.

sequences in 10 insect orders and found a variation in size (673 bp - 690 bp), resulting in between 226 and 229 amino acids. Internal insertions and deletions were observed, but variations length were more frequent in at

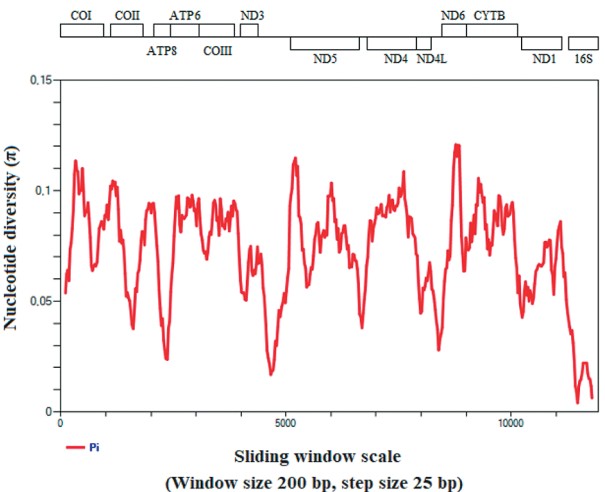

Fig. 6: nucleotide diversity of mitochondrial partial genome of *Culex* (*Mel*.).

or near the 3' end of gene. In the present study, all sequenced species showed deletion of a base near the 3' end of the *ND5* gene, which resulted in the formation of a stop codon and, consequently, a lower number of amino acids. Little is known about the mitochondrial genes of *Culex* (*Mel*.), and therefore, further studies need to be carried out to verify whether this variation can (1) be characteristic of the studied Groups (Atratus and Educator), of the Melanoconion Section or of the subgenus *Melanoconion*, (2) have modified the functions and efficacy of the *ND5* gene.

Partial stop codons (T__) present in *COX1* and *COX2* genes in this study were observed in others *Culex* species and insects.[36,37,38] The presence of these codons is common in mitochondrial PCGs, which are added with adenine by polyadenylation.[39] All species showed positive values for AT-skew and negative values for GC-skew. This result is in agreement with other studies with *Culex*, *Haemagogus*, and *Anopheles*.[20,40,41]

Fig. 7: overview phylogeny of Culicidae made using translations from nine mitochondrial protein coding genes, analysed with Phylobayes MPI V1.9, using two runs. The maxdiff convergence diagnostic was 0.08, showing good topological convergence. However, there was evidence for compositional heterogeneity over the taxa (p = 0.0).

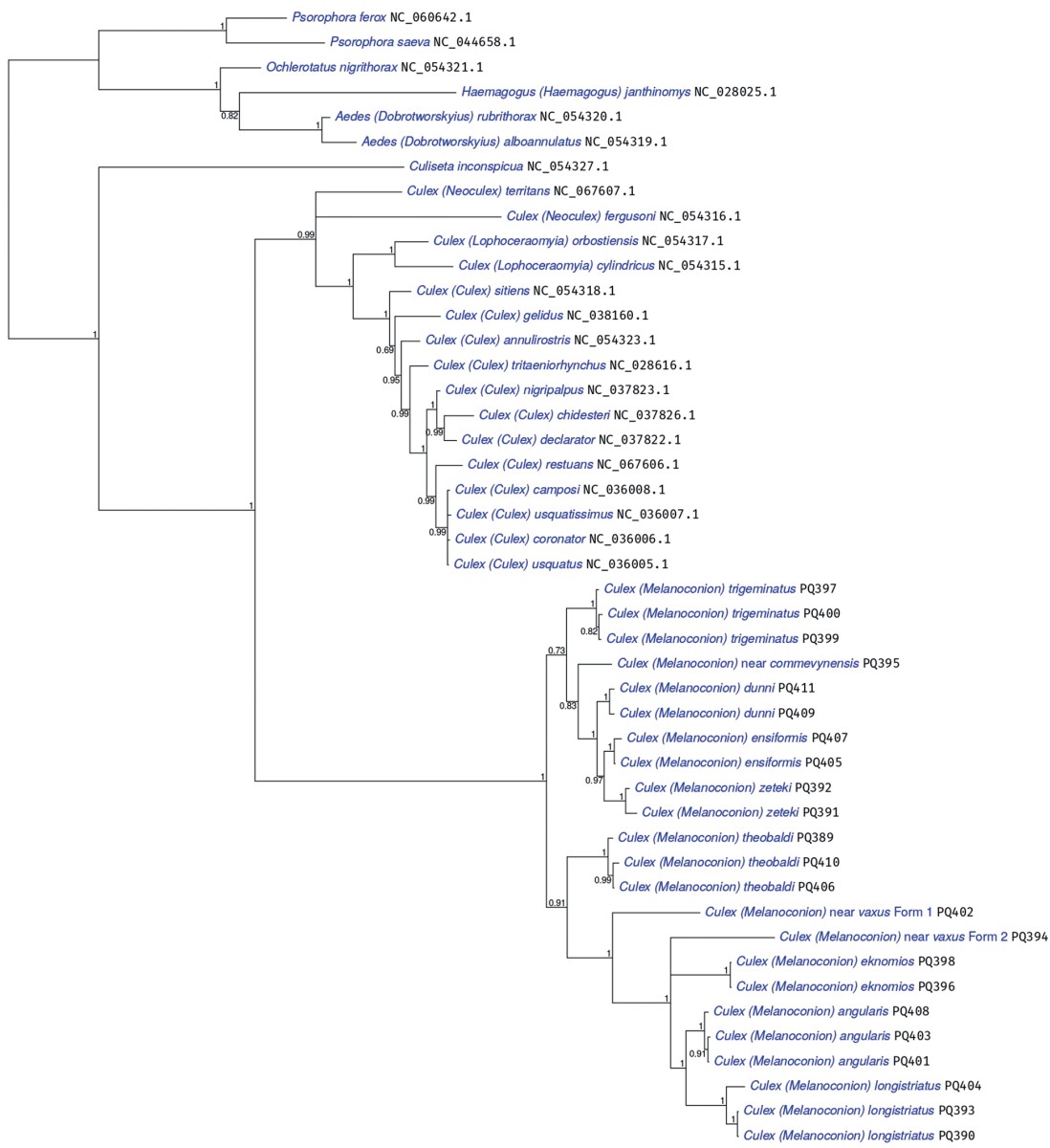

Fig. 8: phylogeny of the *Melanoconion* subgenus, using translations from nine mitochondrial protein coding genes, analysed with Phylobayes MPI V1.9, using two runs. The maxdiff convergence diagnostic was 0.07, showing good topological convergence. Compared to the Culicidae analysis shown in Fig. 7 there was less evidence for compositional heterogeneity across taxa here in these closely-related sequences (p = 0.3, 0.5).

The importance of mitochondrial DNA in the respiratory chain process is well known, as is the high mutational pressure on it in metazoans.[42] Previous studies have reported positive selection (Ka/Ks > 1) in insect mitochondrial PCGs, highlighting its role in adaptation to different environments. Analysis of the mitochondrial genomes of *Anopheles stephensi* and *Anopheles dirus* revealed evidence of positive selection in the *ND2*, *ND4*, and *ND6* genes.[43] Similarly, positive selection was detected in the *ATP8* gene (Ka/Ks = 1.65) among hemipteran species.[44] In addition, seven of the 13 mitochondrial PCGs in flying grasshopper lineages appear to be under positive selection, which may have facilitated adaptation to the high energy demands of sustained flight during periods of atmospheric oxygen reduction.

[45] In the current study, of the 12 mitochondrial PCGs, 11 presented Ka/Ks < 1 (negative selection), as observed in other studies with culicids.[20,36,41] The *ATP8* gene presented Ka/Ks > 1 only in the pairwise sequence comparisons *Cx. longistriatus* (PQ404, PQ390, PQ393) x *Cx. angularis* (PQ401, PQ403, PQ408). These specimens were recovered as sister species in the Bayesian topology (Fig. 8) and further studies need to be carried out to verify whether that selection was fundamental for the adaptation of the species to the environment.

Protein coding and non-coding sequences are used for studies involving *Culex* (*Mel.*) species. Torres-Gutierrez et al.[19] used *COX1* and nuclear genes to infer the phylogeny of species from the Spissipes and Melanononion Sections of *Culex* (*Mel.*) and verified that the

phylogenetic signal of these genes is greater when analysed together than separately. Navarro and Weaver[46] suggested a cryptic species similar to *Culex pedroi* by analysis of the internal transcribed spacer 2 (ITS2) region. In the present study, Bayesian analysis showed that the sequence of the analysed PCGs is highly informative for the separation not only of the Educator and Atratus Groups, but also for species of the others subgenera and genera. Furthermore, this analysis suggested the presence of three putative species, one most morphologically similar to *Cx. commevynensis* and the others to *Cx. vaxus*. The sequences generated may contribute to future studies of the *Melanoconion* subgenus of *Culex*.

Thus, *Cx. dunni*, *Cx. ensiformis*, *Cx. theobaldi*, *Cx. trigeminatus*, *Cx. eknomios*, *Cx. zeteki*, *Cx.* near *commevynensis*, *Cx. longistriatus*, *Cx. angularis*, and *Cx.* near *vaxus* showed the same pattern of PCGs, tRNA, and rRNA as species of the same genus. The PCGs used in the phylogenetic analyses provided good support for monophyletic clades. Future molecular studies will be needed to confirm whether the size difference in the ND5 gene is characteristic of the subgenus *Melanoconion*, and morphological studies will be essential for the analysis of the possible putative *Culex* species.

## AUTHORS' CONTRIBUTION

Conceptualisation - ILRS and MAMS; methodology - TMPO, PGF, ILRS and MAMS; validation, formal analysis and visualisation - TMPO and PGF; investigation, resources, supervision, project administration and funding acquisition - MAMS; data curation and writing-original draft preparation - TMPO; writing-review and editing - PGF, ILRS and MAMS. All authors have read and agreed to the published version of the manuscript. The authors declare no conflicts of interest.

## DATA AVAILABILITY

Sequences of the mitochondrial genomes generated in this study were deposited in Genbank (accession numbers: PV662084-PV662106). Male genitalia of the following samples TO1-9, PI4-100, IG20-5, MS06-100, ES18-109, SP68-35, and MG08-102 were deposited in the Coleção Entomológica de Referência, Faculdade de Saúde Pública, Universidade de São Paulo, with access numbers E-15493, E-14896, E-15492, E-15485, E-15453, E-15446, and E-15445, respectively.

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

# OPEN PEER REVIEW

Memórias do IOC thanks the anonymous reviewers for their contribution to the peer review of this work.

## FIRST REVIEW ROUND

REVIEWERS' COMMENTS

### REVIEWER #1

This paper presents the results of phylogenetic analyses mitogenome sequence data for two groups of species of Culex (Melanoconion), a group of neotropical mosquitoes that includes important vectors of zoonotic diseases. The phylogeny is well resolved, the methods are sound, and the paper is well written overall. Two important points need to be addressed.

1. The Introduction should provide more background on what is previously known about the phylogeny of this group. This is needed to indicate what is new about the current study and why a phylogeny of these particular groups is needed.

2. It is not clear from the Methods whether the DNA extraction was destructive or non-destructive. Were the exoskeletons of extracted specimens saved as vouchers? Were additional voucher specimens of each of the sampled species saved in a research collection that is accessible to other researchers? This is important particularly in cases where species of the group are difficult to identify based on morphology and/or some of the included species were not confidently identified (e.g., "near vaxus" 1, 2, 3, and 4).

### REVIEWER #2

The manuscript describes the partial mitogenomes of 23 specimens of Culex, contributing to the community with more genetic information on those species. Here are some suggestions to improve the work. The abstract is well-written.

In lines 70-74, I wonder why a study would aim to obtain only a fragment/partial mitogenome, missing one PCG. Maybe you should rephrase the aims without already imposing the limitations.

I wonder why the authors opted for amplification of a region that would only allow for obtaining a 12Kb fragment, instead of doing a low coverage WGS and having the possibility of obtaining a circular mitogenome when applying tools such as OrganPipe or NovoPlasty.

Line 109: Authors should describe the parameters used in Geneious to perform the assembly and the refseq used in Mitos 2 for annotation.

Line 115: Adopt whether "nucleotide divergence" or "nucleotide diversity"

The session Phylogenetic analysis (line 127) lacks more details. Were the genes concatenated? Was a supermatrix adopted or a supertree?

Line 130: Parameters used in Phylobayes must be described.

Line 139: I think that a Figure showing the identified Cluex species would add to the manuscript;

Line 148: Authors should state the missing PCG, tRNA, and rRNA;

Line 164: Authors should investigate if the internal stop codon could not be an error, but an RNA exception translation that would be corrected in the transcription step; Edit: Fortunately, the authors describe that this was verified. Maybe authors should rewrite the sentences to make it clear that this was verified.

The quality of the figures must be better

The sentence in lines 196-198 is misplaced, apparently.

The last paragraph (lines 274-280) seems too weak and repetitive. It must be improved.

Accession numbers must be in the manuscript

AUTHORS' RESPONSE TO THE REVIEWERS

Manuscript Number: MIOC-2025-0125

Title: "Mitochondrial genomes and phylogeny of Atratus and Educator Group species of the Melanoconion Section of Culex (Melanoconion) (Diptera: Culicidae)"

Authors: Tatiane M. P. Oliveira, Peter G. Foster, Ivy L. R. Sá and Maria Anice M. Sallum

Dear Editor-in-Chief

We would like to thank the editor and reviewers for taking the time to reading and suggesting changes to the paper. We performed modifications in the revised version of the manuscript, as suggested by reviewers. We hope editor will find the manuscript revised appropriate for publication.

Sincerely,
Tatiane Marques Porangaba de Oliveira
Departamento de Epidemiologia, Faculdade de Saúde Pública, Universidade de São Paulo
Av. Doutor Arnaldo 715, São Paulo, SP 01246-904, Brazil
+55 11 3061-7876
porangaba@usp.br

Before responding to the comments, I would like to inform the reviewers and the editor that the male genitalia slides were re-examined by a renowned and experienced taxonomist, who concluded that the specimens, previously referred to here as Culex near vaxus 1 and Culex near vaxus 2, are Culex longistriatus and Culex angularis, respectively. The misidentification occurred due to the morphology of these species being very similar. Therefore, the necessary changes were made to the tables and images.

Reviewer 1

1. The Introduction should provide more background on what is previously known about the phylogeny of this group. This is needed to indicate what is new about the current study and why a phylogeny of these particular groups is needed.

Response: Thank you for this recommendation. The following information has been added to the introduction:

"Torres-Gutierrez and Sallum(8) updated the catalog of the subgenus Melanoconion published in 1992 by Pecor et al.(9) In the new catalog, this subgenus comprises two sections, 21 groups, 23 subgroups, and 160 valid species. Recently, revisions were carried out on the Atratus and Educator Groups of the Melanoconion Section of the subgenus Melanoconion resulting in the description of eight new species and removal of five species from synonymy.(10,11)" (lines 44-49)

"Demari-Silva et al.(18) used a fragment of the COI gene to establish phylogenetic relationships among 17 species of the genus Culex. Although the results corroborate the monophyly of the subgenus Melanoconion, the authors suggested confirmation through further studies with nuclear genes and a greater number of samples, including species from the Pilosus Group and Spissipes Section. Torres-Gutierrez et al.(19) verified the monophyly of the Melanoconion and Spissipes Sections through phylogenetic analyses with nuclear and mitochondrial genes. Although the results have been consistent with most of the morphological classification of Spissipes Section, the same was not true for Melanoconion Section. Only the Atratus and Pilosus Groups were monophyletic. Therefore, the authors report that the results for Melanoconion Section were inconclusive due to limited taxon representation and suggest future investigations with greater representation.

Mitochondrial genome has also been used to phylogeny in Culicidae.(20,21) Demari-Silva et al.(20) performed phylogenetic analysis with mitochondrial protein-coding genes using four of the six species of the Coronator Group of the subgenus Culex and observed the monophyly of this Group, corroborating with the morphological hypothesis." (lines 68-83)

2. It is not clear from the Methods whether the DNA extraction was destructive or non-destructive. Were the exoskeletons of extracted specimens saved as vouchers? Were additional voucher specimens of each of the sampled species saved in a research collection that is accessible to other researchers? This is important particularly in cases where species of the group are difficult to identify based on morphology and/or some of the included species were not confidently identified (e.g., "near vaxus" 1, 2, 3, and 4).

Response: Thank you for this observation. Male genitalia were removed from whole mosquitoes prior to genomic DNA extraction. Subsequently, the male genitalia of each specimen were dissected and slide-mounted under a coverslip, and some preparations were deposited in the Coleção Entomológica de Referência of the Faculdade de Saúde Pública – Universidade de São Paulo.

In the new version of the manuscript it reads as follows: "Male genitalia were dissected and mounted on microscope slide, covered with fine coverslip, and deposited in the Coleção Entomológica de Referência, Faculdade de Saúde Pública, Universidade de São Paulo. Genomic DNA from whole mosquitoes was individually extracted using the Qiagen DNeasy Blood & Tissue Kit (Qiagen), following the manufacturer's instructions. The extracted DNA was stored at -80 °C as part of the frozen entomological collection of the Faculdade de Saúde Pública, Universidade de São Paulo, Brazil" (lines 105-111)

"Data Availability Statement: Sequences of the mitochondrial genomes generated in this study were deposited in Genbank (accession numbers: PV662084-PV662106). Male genitalia of the following samples TO1-9, PI4-100, IG20-5, MS06-100, ES18-109, SP68-35, and MG08-102 were deposited in the Coleção Entomológica de Referência, Faculdade de Saúde Pública, Universidade de São Paulo, with access numbers E-15493, E-14896, E-15492, E-15485, E-15453, E-15446, and E-15445, respectively."(lines 333-339)

Reviewer 2

In lines 70-74, I wonder why a study would aim to obtain only a fragment/partial mitogenome, missing one PCG. Maybe you should rephrase the aims without already imposing the limitations.

Response: Thank you for this observation. We rephrase the aims and now it reads:

"Thus, this study aims to: (1) obtain, describe, and analyze mitochondrial protein coding genes from different species of the Educator and Atratus Groups of the subgenus Melanoconion; (2) verify the phylogenetic relationships of both Groups." (lines 94-97)

I wonder why the authors opted for amplification of a region that would only allow for obtaining a 12Kb fragment, instead of doing a low coverage WGS and having the possibility of obtaining a circular mitogenome when applying tools such as OrganPipe or NovoPlasty.

Response: The initial idea was to obtain the whole mitochondrial genome by performing single long-range PCR, according to the protocol used in Foster et al. (2017) (https://doi.org/10.1098/rsos.170758). However, the success of this PCR varied between specimens. As 100% success was obtained with PCR performed with primers LCO1490 and 16Sa (~12Kb) and the amplified fragment presented 12 of 13 mitochondrial protein-coding genes, it was decided to sequence this fragment.

Line 109: Authors should describe the parameters used in Geneious to perform the assembly and the refseq used in Mitos 2 for annotation.

Response: Thank you for this observation. Default parameters were used in Geneious. We added more information and now it reads:

"Mapping to reference method was used to genome assembly. Geneious Prime 2023.2.1 (https://www.geneious.com) was used for paired-end reads assembly using default parameters, mapper Geneious method and Culex quinquefasciatus mitochondrial genome sequence (Genbank accession NC_014574) as reference. Annotation of the genes was performed using MITOS 2 Web Server(25) with invertebrate genetic code and confirmed manually with the alignment of each gene with Culex sequences available in Genbank (Genbank accessions NC_036006 and NC_014574)." (lines 131-137)

Line 115: Adopt whether "nucleotide divergence" or "nucleotide diversity"

Response: Thank you for this comment. We adopted "nucleotide diversity".

The session Phylogenetic analysis (line 127) lacks more details. Were the genes concatenated? Was a supermatrix adopted or a supertree?

Response: Thank you for this observation. The following information has been added to session Phylogenetic analysis:

"An overview phylogeny of Culicidae was made using translations from samples of Culicidae species and the newly-sequenced Melanoconion species. Complete mitochondrial genomes from Culicidae were obtained from Genbank, which included 21 genera with at least one mitogenome. Two mitogenomes were sampled randomly from each such genus (with only one from Orthopodomyia, as that was all that was available), from which translations of protein coding genes were obtained. New Melanoconion sequences PQ389 and PQ409 from this study were added to the other Culicidae sequences. Alignments of the translations were made using Clustalo,(28) but alignments for ATP8, ND3, and ND4L were too short (lengths 54, 118, and 99, respectively) and were not used. A concatenated alignment of the remaining nine translations was made. There were five pairs of sequences that were identical; one of each was removed from the alignment. The alignment of the remaining 38 sequences, with a length of 3137 characters, was analysed with Phylobayes MPI V1.9, using two runs, using the CAT+GTR+G(4) model.(29) A phylogeny of the Melanoconion subgenus was made in a similar way to the Culicidae phylogeny described above. Alignments were made using protein-coding gene translations of the 23 new Melanoconion sequences from this study, to which were added Culex and other near outgroup sequences. This alignment of 46 sequences, of length 3127 AAs, was analysed as described above for Culicidae, using Phylobayes MPI V1.9 using the CAT+GTR+G(8) model." (lines 156-175)

The figures 7 and 8 captions were also changed to:

"Figure 7. Overview phylogeny of Culicidae made using translations from 9 mitochondrial protein coding genes, analysed with Phylobayes MPI V1.9, using two runs. The maxdiff convergence diagnostic was 0.08, showing good topological convergence. However, there was evidence for compositional heterogeneity over the taxa (P=0.0).

Figure 8. Phylogeny of the Melanoconion subgenus, using translations from 9 mitochondrial protein coding genes, analysed with Phylobayes MPI V1.9, using two runs. The maxdiff convergence diagnostic was 0.07, showing good topological convergence. Compared to the Culicidae analysis shown in Figure 7 there was less evidence for compositional heterogeneity across taxa here in these closely-related sequences (P=0.3, 0.5)." (lines 539-549)

Line 130: Parameters used in Phylobayes must be described.

Response: Thank you for this comment. Parameters used in Phylobayes were added according to sentence above.

Line 139: I think that a Figure showing the identified Cluex species would add to the manuscript;

Response: Thank you for this comment. Because the objective of this study was to obtain and describe portions of the mitochondrial genome and to perform phylogenetic analyses of the sampled species, we do not see how additional images would enhance interpretation of the results. Furthermore, (1) morphological identifications were made using the keys of Sa et al. (2020) and Rodrigues de Sa et al. (2022), which are readily available in the literature and are comprehensive and well-illustrated; and (2) some slides of male genitalia were deposited in the Coleção de Entomologia de Saúde Pública and are accessible to researchers and students. We added more information and now it reads:

"Data Availability Statement: Sequences of the mitochondrial genomes generated in this study were deposited in Genbank (accession numbers: PV662084-PV662106). Male genitalia of the following samples TO1-9, PI4-100,

IG20-5, MS06-100, ES18-109, SP68-35, and MG08-102 were deposited in the Coleção Entomológica de Referência, Faculdade de Saúde Pública, Universidade de São Paulo, with access numbers E-15493, E-14896, E-15492, E-15485, E-15453, E-15446, and E-15445, respectively."(lines 333-339)

Line 148: Authors should state the missing PCG, tRNA, and rRNA;

Response: Thank you for this recommendation. We added the information below:

"The following nine mitochondrial genes were not sequenced: ND2 (PCG); trnI, trnQ,trnM, trnW, trnC, trnY, and trnV (tRNA); rrnS (rRNA)." (lines 199-201)

Line 164: Authors should investigate if the internal stop codon could not be an error, but an RNA exception translation that would be corrected in the transcription step; Edit: Fortunately, the authors describe that this was verified. Maybe authors should rewrite the sentences to make it clear that this was verified.

Response: Thank you for this comment. We rewrite the sentences and now it reads:

"The ND5 gene of the new sequenced samples presented one less base, which generated a stop codon and, consequently, a shorter length compared to the NC_014574 sequence (Figure 2). Because of the absence of this base occurred in a polyT region, Sanger sequencing was performed to verify whether this difference could have been a result of the assembly of the reads. So, the new sequences were aligned and primers in the ND3 and ND5 regions were designed (Primer F: TTGGACTTTATCATGAATG (ND3 region) and Primer R: GATCAAGGGTGAAGTGAA (ND5 region)) to amplify the fragment containing the polyT region mentioned above. Amplification and sequencing of this fragment (~622 bp) was performed in the following samples: PQ401 (IG20-5), PQ403 (IG20-6), and PQ408 (SP-Ciduni-02). The result of the alignment of the ND5 fragment between Sanger sequencing and Illumina sequencing can be seen in Figure S1, confirming the absence of the base." (lines 210-221)

The quality of the figures must be better

Response: Thank you for this recommendation. The Figures have had their quality improved.

The sentence in lines 196-198 is misplaced, apparently.

Response: Thank you for this observation. We modified the sentence and now it reads:

"The 12kb fragment from the new Melanoconion sequences from this study contained COX1 (partial), COX2, ATP8, ATP6, COX3, ND3, ND5, ND4, ND4L, ND6, CYTB, and ND1. However, ATP8, ND3, and ND4L were not used because they were, with lengths 53, 117, and 98 AAs respectively, deemed too short." (lines 244-248)

The last paragraph (lines 274-280) seems too weak and repetitive. It must be improved.

Response: Thank you for this comment. We reewrited the text and now it reads:

"Thus, Culex dunni, Cx. ensiformis, Cx. theobaldi, Cx. trigeminatus, Cx. eknomios, Cx. zeteki, Cx. near commevynenis, Cx. longistriatus, Cx. angularis, and Cx. near vaxus showed the same pattern of PCGs, tRNA, and rRNA as species of the same genus. The PCGs used in the phylogenetic analyses provided good support for monophyletic clades. Future molecular studies will be needed to confirm whether the size difference in the ND5 gene is characteristic of the subgenus Melanoconion, and morphological studies will be essential for the analysis of the possible putative Culex species." (lines 323-329)

Accession numbers must be in the manuscript

Response: Thank you for this observation. We added the accession numbers in data availability statement and now it reads:

"Data Availability Statement: Sequences of the mitochondrial genomes generated in this study were deposited in Genbank (accession numbers: PV662084-PV662106)." (lines 333-334).

## SECOND REVIEW ROUND

REVIEWERS' COMMENTS

### REVIEWER #1

Most suggestions I made were accepted, and for those that weren't, the authors provided justified explanations.

### REVIEWER #2

No comments.

