## [Reviewer Report · FIRST REVIEW ROUND - REVIEWERS COMMENTS]

## REVIEWER #1

This paper presents the results of phylogenetic analyses mitogenome sequence data for two groups of species of *Culex* (*Melanoconion*), a group of neotropical mosquitoes that includes important vectors of zoonotic diseases. The phylogeny is well resolved, the methods are sound, and the paper is well written overall. Two important points need to be addressed.

1. The Introduction should provide more background on what is previously known about the phylogeny of this group. This is needed to indicate what is new about the current study and why a phylogeny of these particular groups is needed.

2. It is not clear from the Methods whether the DNA extraction was destructive or non-destructive. Were the exoskeletons of extracted specimens saved as vouchers? Were additional voucher specimens of each of the sampled species saved in a research collection that is accessible to other researchers? This is important particularly in cases where species of the group are difficult to identify based on morphology and/or some of the included species were not confidently identified (e.g., "near *vaxus*" 1, 2, 3, and 4).

## REVIEWER #2

The manuscript describes the partial mitogenomes of 23 specimens of *Culex*, contributing to the community with more genetic information on those species. Here are some suggestions to improve the work. The abstract is well-written.

In lines 70-74, I wonder why a study would aim to obtain only a fragment/partial mitogenome, missing one PCG. Maybe you should rephrase the aims without already imposing the limitations.

I wonder why the authors opted for amplification of a region that would only allow for obtaining a 12Kb fragment, instead of doing a low coverage WGS and having the possibility of obtaining a circular mitogenome when applying tools such as OrganPipe or NovoPlasty.

Line 109: Authors should describe the parameters used in Geneious to perform the assembly and the refseq used in Mitos 2 for annotation.

Line 115: Adopt whether "nucleotide divergence" or "nucleotide diversity"

The session Phylogenetic analysis (line 127) lacks more details. Were the genes concatenated? Was a supermatrix adopted or a supertree?

Line 130: Parameters used in Phylobayes must be described.

Line 139: I think that a Figure showing the identified *Cluex* species would add to the manuscript;

Line 148: Authors should state the missing PCG, tRNA, and rRNA;

Line 164: Authors should investigate if the internal stop codon could not be an error, but an RNA exception translation that would be corrected in the transcription step; Edit: Fortunately, the authors describe that this was verified. Maybe authors should rewrite the sentences to make it clear that this was verified.

The quality of the figures must be better

The sentence in lines 196-198 is misplaced, apparently.

The last paragraph (lines 274-280) seems too weak and repetitive. It must be improved.

Accession numbers must be in the manuscript.

## AUTHORS' RESPONSE TO THE REVIEWERS

**Manuscript Number:** MIOC-2025-0125

**Title:** "Mitochondrial genomes and phylogeny of Atratus and Educator Group species of the Melanoconion Section of *Culex* (*Melanoconion*) (Diptera: Culicidae)"

**Authors:** Tatiane M. P. Oliveira, Peter G. Foster, Ivy L. R. Sá and Maria Anice M. Sallum

Dear Editor-in-Chief

We would like to thank the editor and reviewers for taking the time to reading and suggesting changes to the paper. We performed modifications in the revised version of the manuscript, as suggested by reviewers. We hope editor will find the manuscript revised appropriate for publication.

Sincerely,

Tatiane Marques Porangaba de Oliveira

Departamento de Epidemiologia, Faculdade de Saúde Pública, Universidade de São Paulo

Av. Doutor Arnaldo 715, São Paulo, SP 01246-904, Brazil

+55 11 3061-7876

porangaba@usp.br

Before responding to the comments, I would like to inform the reviewers and the editor that the male genitalia slides were re-examined by a renowned and experienced taxonomist, who concluded that the specimens, previously referred to here as *Culex* near *vaxus* 1 and *Culex* near *vaxus* 2, are *Culex longistriatus* and *Culex angularis*, respectively. The misidentification occurred due to the morphology of these species being very similar. Therefore, the necessary changes were made to the tables and images.

## Reviewer 1

**1. The Introduction should provide more background on what is previously known about the phylogeny of this group. This is needed to indicate what is new about the current study and why a phylogeny of these particular groups is needed.**

*Response:* Thank you for this recommendation. The following information has been added to the introduction:

"Torres-Gutierrez and Sallum(8) updated the catalog of the subgenus *Melanoconion* published in 1992 by Pecor et al.(9) In the new catalog, this subgenus comprises two sections, 21 groups, 23 subgroups, and 160 valid species. Recently, revisions were carried out on the Atratus and Educator Groups of the Melanoconion Section of the subgenus *Melanoconion* resulting in the description of eight new species and removal of five species from synonymy.(10,11)" (lines 44-49)

"Demari-Silva et al.(18) used a fragment of the COI gene to establish phylogenetic relationships among 17 species of the genus *Culex*. Although the results corroborate the monophyly of the subgenus *Melanoconion*, the authors suggested confirmation through further studies with nuclear genes and a greater number of samples, including species from the Pilosus Group and Spissipes Section. Torres-Gutierrez et al.(19) verified the monophyly of the Melanoconion and Spissipes Sections through phylogenetic analyses with nuclear and mitochondrial genes. Although the results have been consistent with most of the morphological classification of Spissipes Section, the same was not true for Melanoconion Section. Only the Atratus and Pilosus Groups were monophyletic. Therefore, the authors report that the results for Melanoconion Section were inconclusive due to limited taxon representation and suggest future investigations with greater representation.

Mitochondrial genome has also been used to phylogeny in Culicidae.(20,21) Demari-Silva et al.(20) performed phylogenetic analysis with mitochondrial protein-coding genes using four of the six species of the Coronator Group of the subgenus *Culex* and observed the monophyly of this Group, corroborating with the morphological hypothesis." (lines 68-83)

**2. It is not clear from the Methods whether the DNA extraction was destructive or non-destructive. Were the exoskeletons of extracted specimens saved as vouchers? Were additional voucher specimens of each of the sampled species saved in a research collection that is accessible to other researchers? This is important particularly in cases where species of the group are difficult to identify based on morphology and/or some of the included species were not confidently identified (e.g., "near vaxus" 1, 2, 3, and 4).**

*Response:* Thank you for this observation. Male genitalia were removed from whole mosquitoes prior to genomic DNA extraction. Subsequently, the male genitalia of each specimen were dissected and slide-mounted under a coverslip, and some preparations were deposited in the Coleção Entomológica de Referência of the Faculdade de Saúde Pública – Universidade de São Paulo.

In the new version of the manuscript it reads as follows:"Male genitalia were dissected and mounted on microscope slide, covered with fine coverslip, and deposited in the Coleção Entomológica de Referência, Faculdade de Saúde Pública, Universidade de São Paulo. Genomic DNA from whole mosquitoes was individually extracted using the Qiagen DNeasy Blood & Tissue Kit (Qiagen), following the manufacturer's instructions. The extracted DNA was stored at -80 °C as part of the frozen entomological collection of the Faculdade de Saúde Pública, Universidade de São Paulo, Brazil" (lines 105-111)

"Data Availability Statement: Sequences of the mitochondrial genomes generated in this study were deposited in Genbank (accession numbers: PV662084-PV662106). Male genitalia of the following samples TO1-9, PI4-100, IG20-5, MS06-100, ES18-109, SP68-35, and MG08-102 were deposited in the Coleção Entomológica de Referência, Faculdade de Saúde Pública, Universidade de São Paulo, with access numbers E-15493, E-14896, E-15492, E-15485, E-15453, E-15446, and E-15445, respectively."(lines 333-339)

## Reviewer 2

**In lines 70-74, I wonder why a study would aim to obtain only a fragment/partial mitogenome, missing one PCG. Maybe you should rephrase the aims without already imposing the limitations.**

*Response:* Thank you for this observation. We rephrase the aims and now it reads:

"Thus, this study aims to: (1) obtain, describe, and analyze mitochondrial protein coding genes from different species of the Educator and Atratus Groups of the subgenus *Melanoconion*; (2) verify the phylogenetic relationships of both Groups." (lines 94-97)

**I wonder why the authors opted for amplification of a region that would only allow for obtaining a 12Kb fragment, instead of doing a low coverage WGS and having the possibility of obtaining a circular mitogenome when applying tools such as OrganPipe or NovoPlasty.**

*Response:* The initial idea was to obtain the whole mitochondrial genome by performing single long-range PCR, according to the protocol used in Foster et al. (2017) (https://doi.org/10.1098/rsos.170758). However, the success of this PCR varied between specimens. As 100% success was obtained with PCR performed with primers LCO1490 and 16Sa (~12Kb) and the amplified fragment presented 12 of 13 mitochondrial protein-coding genes, it was decided to sequence this fragment.

**Line 109: Authors should describe the parameters used in Geneious to perform the assembly and the refseq used in Mitos 2 for annotation.**

*Response:* Thank you for this observation. Default parameters were used in Geneious. We added more information and now it reads:

"Mapping to reference method was used to genome assembly. Geneious Prime 2023.2.1 (https://www.geneious.com) was used for paired-end reads assembly using default parameters, mapper Geneious method and *Culex quinquefasciatus* mitochondrial genome sequence (Genbank accession NC_014574) as reference. Annotation of the genes was performed using MITOS 2 Web Server(25) with invertebrate genetic code and confirmed manually with the alignment of each gene with *Culex* sequences available in Genbank (Genbank accessions NC_036006 and NC_014574)." (lines 131-137)

**Line 115: Adopt whether "nucleotide divergence" or "nucleotide diversity"**

*Response:* Thank you for this comment. We adopted "nucleotide diversity".

**The session Phylogenetic analysis (line 127) lacks more details. Were the genes concatenated? Was a supermatrix adopted or a supertree?**

*Response:* Thank you for this observation. The following information has been added to session Phylogenetic analysis:

"An overview phylogeny of Culicidae was made using translations from samples of Culicidae species and the newly-sequenced *Melanoconion* species. Complete mitochondrial genomes from Culicidae were obtained from Genbank, which included 21 genera with at least one mitogenome. Two mitogenomes were sampled randomly from each such genus (with only one from *Orthopodomyia*, as that was all that was available), from which translations of protein coding genes were obtained. New *Melanoconion* sequences PQ389 and PQ409 from this study were added to the other Culicidae sequences. Alignments of the translations were made using Clustalo,(28) but alignments for ATP8, ND3, and ND4L were too short (lengths 54, 118, and 99, respectively) and were not used. A concatenated alignment of the remaining nine translations was made. There were five pairs of sequences that were identical; one of each was removed from the alignment. The alignment of the remaining 38 sequences, with a length of 3137 characters, was analysed with Phylobayes MPI V1.9, using two runs, using the CAT+GTR+G(4) model.(29) A phylogeny of the *Melanoconion* subgenus was made in a similar way to the Culicidae phylogeny described above. Alignments were made using protein-coding gene translations of the 23 new *Melanoconion* sequences from this study, to which were added *Culex* and other near outgroup sequences. This alignment of 46 sequences, of length 3127 AAs, was analysed as described above for Culicidae, using Phylobayes MPI V1.9 using the CAT+GTR+G(8) model." (lines 156-175)

The figures 7 and 8 captions were also changed to:

"Figure 7. Overview phylogeny of Culicidae made using translations from 9 mitochondrial protein coding genes, analysed with Phylobayes MPI V1.9, using two runs. The maxdiff convergence diagnostic was 0.08, showing good topological convergence. However, there was evidence for compositional heterogeneity over the taxa (P=0.0).

Figure 8. Phylogeny of the *Melanoconion* subgenus, using translations from 9 mitochondrial protein coding genes, analysed with Phylobayes MPI V1.9, using two runs. The maxdiff convergence diagnostic was 0.07, showing good topological convergence. Compared to the Culicidae analysis shown in Figure 7 there was less evidence for compositional heterogeneity across taxa here in these closely-related sequences (P=0.3, 0.5)." (lines 539-549)

**Line 130: Parameters used in Phylobayes must be described.**

*Response:* Thank you for this comment. Parameters used in Phylobayes were added according to sentence above.

**Line 139: I think that a Figure showing the identified Cluex species would add to the manuscript;**

*Response:* Thank you for this comment. Because the objective of this study was to obtain and describe portions of the mitochondrial genome and to perform phylogenetic analyses of the sampled species, we do not see how additional images would enhance interpretation of the results. Furthermore, (1) morphological identifications were made using the keys of Sa et al. (2020) and Rodrigues de Sa et al. (2022), which are readily available in the literature and are comprehensive and well-illustrated; and (2) some slides of male genitalia were deposited in the Coleção de Entomologia de Saúde Pública and are accessible to researchers and students. We added more information and now it reads:

"Data Availability Statement: Sequences of the mitochondrial genomes generated in this study were deposited in Genbank (accession numbers: PV662084-PV662106). Male genitalia of the following samples TO1-9, PI4-100, IG20-5, MS06-100, ES18-109, SP68-35, and MG08-102 were deposited in the Coleção Entomológica de Referência, Faculdade de Saúde Pública, Universidade de São Paulo, with access numbers E-15493, E-14896, E-15492, E-15485, E-15453, E-15446, and E-15445, respectively."(lines 333-339)

**Line 148: Authors should state the missing PCG, tRNA, and rRNA;**

*Response:* Thank you for this recommendation. We added the information below:

"The following nine mitochondrial genes were not sequenced: ND2 (PCG); trnI, trnQ,trnM, trnW, trnC, trnY, and trnV (tRNA); rrnS (rRNA)." (lines 199-201)

**Line 164: Authors should investigate if the internal stop codon could not be an error, but an RNA exception translation that would be corrected in the transcription step; Edit: Fortunately, the authors describe that this was verified. Maybe authors should rewrite the sentences to make it clear that this was verified.**

*Response:* Thank you for this comment. We rewrite the sentences and now it reads:

"The ND5 gene of the new sequenced samples presented one less base, which generated a stop codon and, consequently, a shorter length compared to the NC_014574 sequence (Figure 2). Because of the absence of this base occurred in a polyT region, Sanger sequencing was performed to verify whether this difference could have been a result of the assembly of the reads. So, the new sequences were aligned and primers in the ND3 and ND5 regions were designed (Primer F: TTGGACTTTATCATGAATG (ND3 region) and Primer R: GATCAAGGGTGAAGTGAA (ND5 region)) to amplify the fragment containing the polyT region mentioned above. Amplification and sequencing of this fragment (~622 bp) was performed in the following samples: PQ401 (IG20-5), PQ403 (IG20-6), and PQ408 (SP-Ciduni-02). The result of the alignment of the ND5 fragment between Sanger sequencing and Illumina sequencing can be seen in Figure S1, confirming the absence of the base." (lines 210-221)

**The quality of the figures must be better**

*Response:* Thank you for this recommendation. The Figures have had their quality improved.

**The sentence in lines 196-198 is misplaced, apparently.**

*Response:* Thank you for this observation. We modified the sentence and now it reads:

"The 12kb fragment from the new *Melanoconion* sequences from this study contained COX1 (partial), COX2, ATP8, ATP6, COX3, ND3, ND5, ND4, ND4L, ND6, CYTB, and ND1. However, ATP8, ND3, and ND4L were not used because they were, with lengths 53, 117, and 98 AAs respectively, deemed too short." (lines 244-248)

**The last paragraph (lines 274-280) seems too weak and repetitive. It must be improved.**

*Response:* Thank you for this comment. We reewrited the text and now it reads:

"Thus, *Culex dunni*, *Cx. ensiformis*, *Cx. theobaldi*, *Cx. trigeminatus*, *Cx. eknomios*, *Cx. zeteki*, *Cx.* near *commevynenis*, *Cx. longistriatus*, *Cx. angularis*, and *Cx.* near *vaxus* showed the same pattern of PCGs, tRNA, and rRNA as species of the same genus. The PCGs used in the phylogenetic analyses provided good support for monophyletic clades. Future molecular studies will be needed to confirm whether the size difference in the ND5 gene is characteristic of the subgenus *Melanoconion*, and morphological studies will be essential for the analysis of the possible putative *Culex* species." (lines 323-329)

**Accession numbers must be in the manuscript**

*Response:* Thank you for this observation. We added the accession numbers in data availability statement and now it reads:

"Data Availability Statement: Sequences of the mitochondrial genomes generated in this study were deposited in Genbank (accession numbers: PV662084-PV662106)." (lines 333-334).

---

## [Reviewer Report · REVIEWERS COMMENTS]

## REVIEWER #1

Most suggestions I made were accepted, and for those that weren’t, the authors provided justified explanations.

## REVIEWER #2

No comments.